# Pilot study on the value of Raman spectroscopy in the entity assignment of salivary gland tumors

Till Jasper Meyer[1]*, Elena Gerhard-Hartmann[2], Nina Lodes[3], Agmal Scherzad[1], Rudolf Hagen[1], Maria Steinke[3,4]◔, Stephan Hackenberg[1,5]◔

1 Department of Oto-Rhino-Laryngology, Plastic, Aesthetic & Reconstructive Head and Neck Surgery, University Hospital Würzburg, Würzburg, Germany, 2 Institute of Pathology, University of Würzburg, Würzburg, Germany, 3 Chair of Tissue Engineering and Regenerative Medicine, University Hospital Würzburg, Würzburg, Germany, 4 Fraunhofer Institute for Silicate Research ISC, Würzburg, Germany, 5 Department of Otorhinolaryngology – Head and Neck Surgery, RWTH Aachen University Hospital, Aachen, Germany

◔ These authors contributed equally to this work.
* meyer_t2@ukw.de

**Data Availability Statement:** All relevant data are within the manuscript and its Supporting information files.

## Abstract

### Background

The entity assignment of salivary gland tumors (SGT) based on histomorphology can be challenging. Raman spectroscopy has been applied to analyze differences in the molecular composition of tissues. The aim of this study was to evaluate the suitability of RS for entity assignment in SGT.

### Methods

Raman data were collected in deparaffinized sections of pleomorphic adenomas (PA) and adenoid cystic carcinomas (ACC). Multivariate data and chemometric analysis were completed using the Unscrambler software.

### Results

The Raman spectra detected in ACC samples were mostly assigned to nucleic acids, lipids, and amides. In a principal component-based linear discriminant analysis (LDA) 18 of 20 tumor samples were classified correctly.

### Conclusion

In this proof of concept study, we show that a reliable SGT diagnosis based on LDA algorithm appears possible, despite variations in the entity-specific mean spectra. However, a standardized workflow for tissue sample preparation, measurement setup, and chemometric algorithms is essential to get reliable results.

**Funding:** This work was founded by the Interdisciplinary Centre for Clinical Science (IZKF) of the University of Würzburg, Grant Number Z-2/78 to TJM. The funders had no role in study design, data collection and analysis, decision to publish, or preparation of the manuscript.

**Competing interests:** The authors have declared that no competing interests exist.

**Abbreviations:** ACC, Adenoid cystic carcinoma; FN, false negative; FP, false positive; FNAC, Fine-needle aspiration cytology; FFPE, Formalin-fixed, Paraffin-embedded; FS, Frozen sections; HE, Haematoxylin and eosin; LDA, linear discriminant analysis; PA, pleomorphic adenoma; PC, principal component; PCA, principal component analysis; RS, Raman spectroscopy; SGT, Salivary gland tumors; TN, true negative; TP, true positive.

# Introduction

Tumors of the salivary glands are rare and account for approximately 3 to 6% of all head and neck neoplasms [1]. However, there is a huge variability of benign and malignant salivary gland tumor entities, the current WHO Classification of Head and Neck tumors counts more than 30 different malignant and benign tumor entities [2–4]. The most common tumor type is the salivary gland pleomorphic adenoma (PA), which accounts for roughly 70% of benign epithelial tumors followed by the Warthin's tumor [5]. The adenoid cystic carcinoma (ACC) is the most common malignant tumor in the minor salivary glands, and second most frequent malignancy in the major salivary glands [6]. The most frequent malignancy in the major salivary glands is the mucoepidermoid carcinoma [6].

The huge variety of salivary gland tumor entities makes a fast pre-, intra- and postoperative diagnosis based on cyto- or histomorphological criteria difficult and in many cases even impossible. The uncertainties regarding the pre- and intraoperative determination of dignity and entity makes it impossible to assign the right extension of the surgical therapy in many cases. In benign parotid gland tumors, partial removal of the salivary gland tissue including the entire tumor is preferred. In contrast, malignant lesions require more extensive surgery. Some authors underline the importance of an accurate preoperative assessment of the dignity of SGT in order to improve the quality of the surgical management [7]. However, preoperative diagnosis is often difficult, and the discrimination between different tumor entities is impossible in many cases. Besides ultrasound and magnetic resonance imaging, cytopathological evaluation can be part of the preoperative tumor characterization. Fine-needle aspiration cytology (FNAC) is a well-established but controversially discussed method for preoperative classification of SGT. Low sensitivity of FNAC is probably the most limiting aspect of this minimal-invasive procedure [8]. By FNAC, only single cells can be obtained without the architecture of the tumor invasion front. Thus, the detection of malignancies is challenging especially in low-grade tumor types [9]. Therapy management can be improved significantly by intraoperative frozen section (FS) pathology which proved to be an accurate tool to discriminate salivary gland neoplasms, and to improve decision making during surgery [9, 10]. However, sensitivity of FS varies within the literature between 77% and 93% [10–12].

Consequently, there is a demand for suitable technologies providing quick results to improve the accuracy of salivary gland neoplasm classification. Raman spectra originate of inelastic light scattering of a monochromatic laser light. The inelastic light scattering, also called Raman scattering or Stokes- and anti-Stokes-scattering, is depending on the molecular composition of the tissue [13–15]. The essential components of a Raman spectroscopy (RS) system are a monochromatic laser light source, a filter, which collects the elastic scattered light by being impermeable for light of the same wavelength like the laser light, and a spectrometer (Fig 1). RS is non-invasive, only a small sample size and few measurement steps are necessary to get reliable data on tissue- or even cell-specific molecular characteristics. Using multivariate data analysis, biologically relevant information can be extracted from the spectral data.

RS has previously been used in salivary gland cancer to discriminate PA, Warthin's tumor, and mucoepidermoid carcinoma from normal parotid gland tissue with high accuracy [16, 17]. As opposed to head and neck squamous cell cancer therapy, the most challenging topic in salivary gland cancer cases is usually not tumor margin assessment. More important in salivary gland cancer is the tumor entity determination.

In summary, established pre- and intraoperative diagnostics in SGT therapy like FNAC and FS pathology show insecurities. RS theoretically provides useful properties that might improve accuracy in this context. Thus, the aim of this study was to evaluate a RS-based classification model to discriminate different SGT entities. PA and ACC are the most frequent benign and

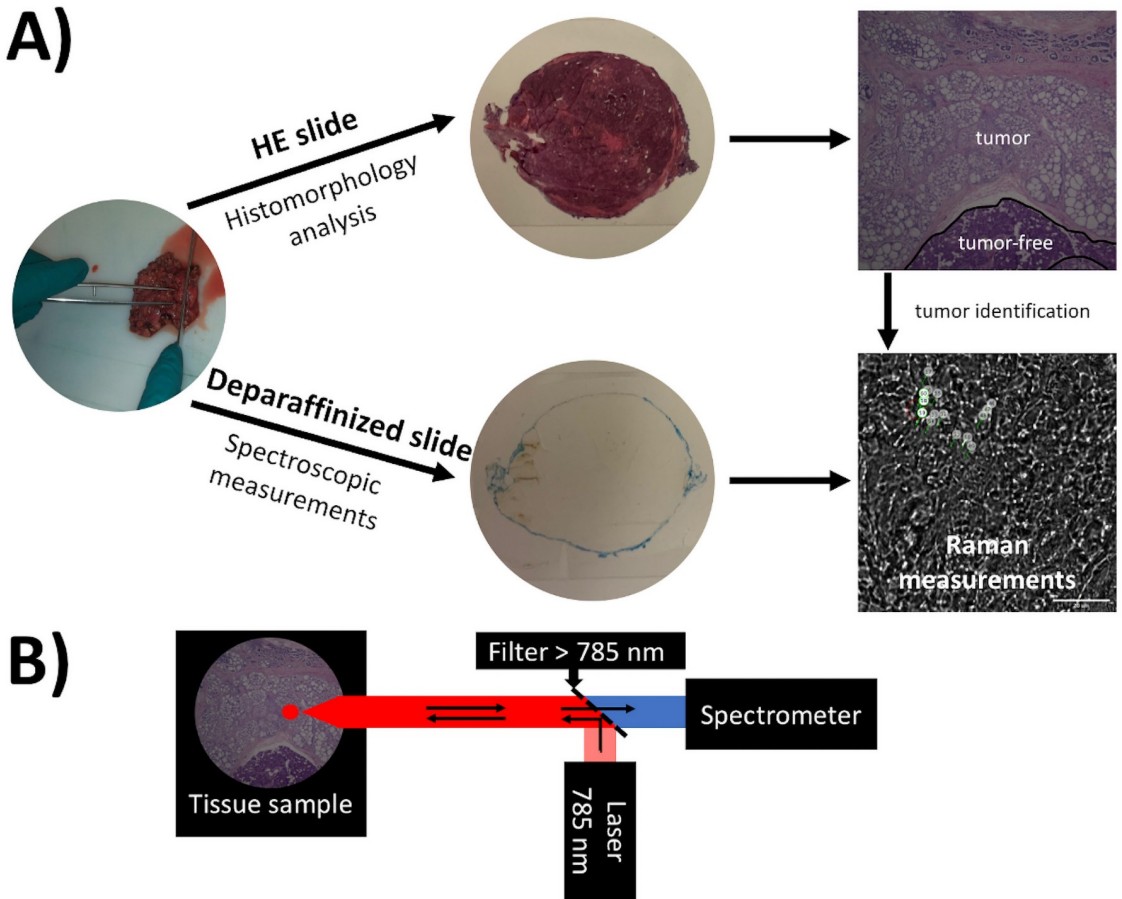

**Fig 1. The workflow of Raman measurements and the Raman spectroscopy setup.** A) HE-stained slides were used for histopathological tumor identification and consecutive deparaffined slides for Raman spectroscopic measurements in the identified tumor regions. B) The Raman spectroscopy setup consists of a 785 nm laser for excitation. Elastic scattered light was removed by a filter that is impermeable for light with a wavelength 785 nm and smaller.

malignant SGT and therefore they were chosen as model entities for benign and malignant SGT.

## Materials and methods

### Tissue selection

Formalin-fixed, paraffin-embedded (FFPE) SGT samples from 20 patients diagnosed with PA and ACC (ten each) were identified from the files of the Institute of Pathology, University of Würzburg. All samples were obtained within primary SGT operation. The histopathological diagnosis was confirmed by two independent, trained pathologists (Table 1).

This study was approved by the institutional ethics committee on human research of the Julius-Maximilians-University Würzburg (vote No. 20170509 01). All experiments were performed according to the Declaration of Helsinki. Informed consent by the patients is not required due to the use of anonymized biomaterial. The biomaterial was anonymized before the authors had access to it.

**Table 1. Anonymized patient information.**

|  | PA | ACC |
|---|---|---|
| **Age range [yrs]** | 43–84 | 19–89 |
| **Gender** |  |  |
| **Female** | 6 | 5 |
| **Male** | 4 | 5 |

## Histologic samples

Parotid tissue samples were fixed in 4% neutral buffered formalin and paraffin-embedded. From each block, we cut 20 μm-thick sequential sections, mounted them on borosilicate cover-slips with a thickness of 0.12 to 0.16 mm (R. Langenbrinck GmbH, Emmendingen, Germany), air-dried them overnight at 37°C, deparaffinized, and rehydrated them. For histologic assessment, we stained consecutive 3 μm thick sections with haematoxylin and eosin (HE) according to a standard protocol. The remaining sections were left unstained, air-dried at 37°C and used for Raman data collection.

## Raman spectroscopy

To collect Raman spectra we used the confocal BioRam® system (CellTool GmbH, Tutzing, Germany) with a non-destructive 785 nm diode laser (Toptica Photonics, Gräfelfing, Germany) combining Raman spectrometry with digital microscopy [18, 19]. The system was calibrated using silicon showing a characteristic Raman peak at 520 cm$^{-1}$. In HE-stained sections we identified tumor-specific areas and took the Raman measurements in the corresponding area of the unstained sequential section (Fig 1).

The Raman laser with a maximal excitation power of 80 mW was focused through a 60x water immersion objective and we took 30 Raman spectroscopy measurements per sample (3 x 10 sec, each). To reflect tumor heterogeneity *in vivo* the measurements were taken randomly in nuclei, cytoplasm, and extracellular matrix. However, tumor-free areas, nerves, muscles, and blood vessels were excluded from measurements. HE-stained samples were evaluated and photographed using the BZ-9000 BIOREVO System (Keyence, Neu-Isenburg, Germany).

## Data pre-treatment and multivariate data analysis

Data were processed applying customized BioRam® software by CellTool [20], including baseline correction. The baseline correction was performed independently for every single sample. The data were transferred to the Unscrambler X 10.3 software (Camo Software, Oslo, Norway). All raw spectra were checked for outliers with a non-biological signature (for example with glass typical peaks) in the line plot. We cropped the spectra to the fingerprint region from 1700–600 cm$^{-1}$, consisting of 249 variables. Mean spectra were calculated first depending on the tumor entity, and second depending on the single patients' samples. The entity specific mean spectra including the standard deviation were analyzed in the line plot for intensity differences and compared to the wavenumber specific biological assignments.

To reduce the complex spectral information, we performed a principal component analysis (PCA) with mean centered data, and seven principle components using the NIPALS algorithm and leverage correction.

For supervised data classification linear discriminant analysis (LDA) using the first six principal component (PC) scores as predictors was performed. For approval of algorithm efficiency, the calculation of the following parameters was performed (true negative (TN); false

negative (FN); true positive (TP); false positive (FP)):

$$Specificity = (TN/TN + FN)$$

$$Sensitivity = TP/(TP + FP)$$

$$Accuracy = (TP + TN)/(TP + TN + FP + FN)$$

$$Matthew\ correlation\ coefficient\ (MCC) = (TP*TN - FP*FN)/\sqrt{((TP + FN)*(TP + FP)*(TN + FP)*(TN + FN))}$$

$$error = FP/(FP/(FP + TN))$$

$$Rigidity = (2*(Accuracy - error)/1 + \ Accuracy - error\ )$$

## Results

### Histologic characterization of HE-stained samples

Histologically, the PA samples showed the typical features of this benign salivary gland neoplasm, with well circumscribed tumors composed of a mixture of ductal epithelial and myoepithelial cells as well as mesenchymal stromal elements in different proportions. In contrast, the HE-stained sections from the ACC samples showed tumors with strongly infiltrative margins and perineural invasion, characteristic for this malignant salivary gland tumor, composed of epithelial/ myoepithelial cells mainly arranged in tubular and cribriform patterns (Fig 2).

### Mean spectra analysis

Mean spectra plots were generated to directly compare Raman spectra intensities of both tumor entities to each other. Visual comparison of the PA and ACC mean spectra revealed differences from 1670–1644, 1456–1428, 1294–1284, 1248–1213, 1207–1193, 1124–1111, 1060–1047, and 1003–989 cm$^{-1}$ with higher Raman spectral intensities in the ACC group, and from 935–914 and 900–888 cm$^{-1}$ with a higher intensities in the PA group (Table 2 and Fig 3). The wavenumber ranges from 1670–1644, 1297–1284, 1248–1213, 1207–1193, and 1124–1111

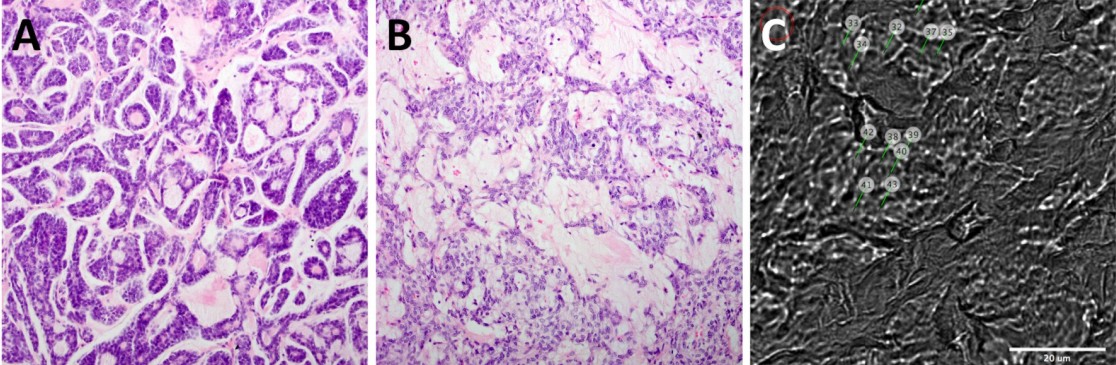

**Fig 2. Histological properties of ACC and PA samples.** Example for HE-stained A) ACC and B) PA tissue samples, imaged in 20-fold magnification. C) shows a representative image as seen in the Raman spectroscope integrated digital microscope with a 60-fold magnification. Single measuring points are indicated.

**Table 2. Entity dependent Raman intensities and biochemical assignments of the visually analyzed mean spectra: The Raman spectra detected in ACC are mostly assigned to nucleic acids, lipids, and amides.** Comparably higher intensities in PA were measured for peaks coding for collagens, and saccharides (based on [14, 15]).

| Wavenumber range [cm⁻¹] | PA [mean intensity] | ACC [mean intensity] | Major biochemical assignments |
|---|---|---|---|
| 1670–1644 | 960 | *1155* | Amide, nucleoid acids |
| 1456–1428 | 1280 | *2960* | Proteins, lipids |
| 1294–1284 | 475 | *845* | Nucleic acid |
| 1248–1213 | 630 | *845* | Amide |
| 1207–1193 | 310 | *450* | Nucleic acids |
| 1124–1111 | 900 | *1130* | RNA, glucose |
| 1060–1047 | 690 | *930* | Lipids, proteins, glycogen |
| 1030 | 240 | *301* | Phenylalanine |
| 1003 | 1490 | *1850* | Phenylalanine |
| 935–914 | *740* | 575 | Proteins, collagen |
| 900–888 | *420* | 230 | Proteins, saccharides |

cm⁻¹ are mainly assigned to nucleic acids, 1456–1428, and 1060–1047 cm⁻¹ to lipids, 1670–1644, 1456–1428, 1248–1213 cm⁻¹ to proteins, and 1003–989 cm⁻¹ to phenylalanine. In contrast, in the wavenumber area from 935–888 cm⁻¹, mainly coding for proteins, collagens, and saccharides, the mean spectral intensity was increased in the PA group. Taken the standard deviation in consideration the most relevant differences between PA and ACC mean spectra appeared to be around the wavenumbers 1290 cm⁻¹ and 900 cm⁻¹ (Fig 3).

To sum up, the Raman mean spectra suggest, that ACC samples contain a higher amount of DNA, RNA, and lipids and a lower amount of collagen, and saccharides in comparison to PA samples.

## Raman data analysis by PCA

PCA with all acquired single spectra revealed that there was a clustering in dependence of the different tumor samples visible (S1 Fig).

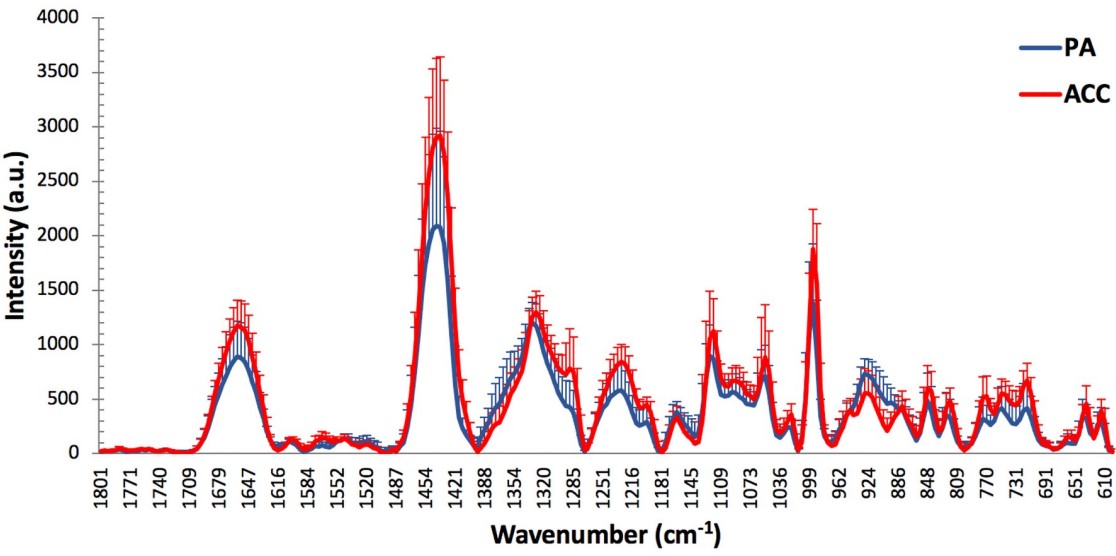

**Fig 3. Entity dependent mean spectra: Mean Raman spectra of PA and ACC samples: The spectral intensity of ACC samples is mostly higher compared to PA samples.** Only in the wavenumber area 935–888 cm⁻¹ the intensity measured in PA is higher.

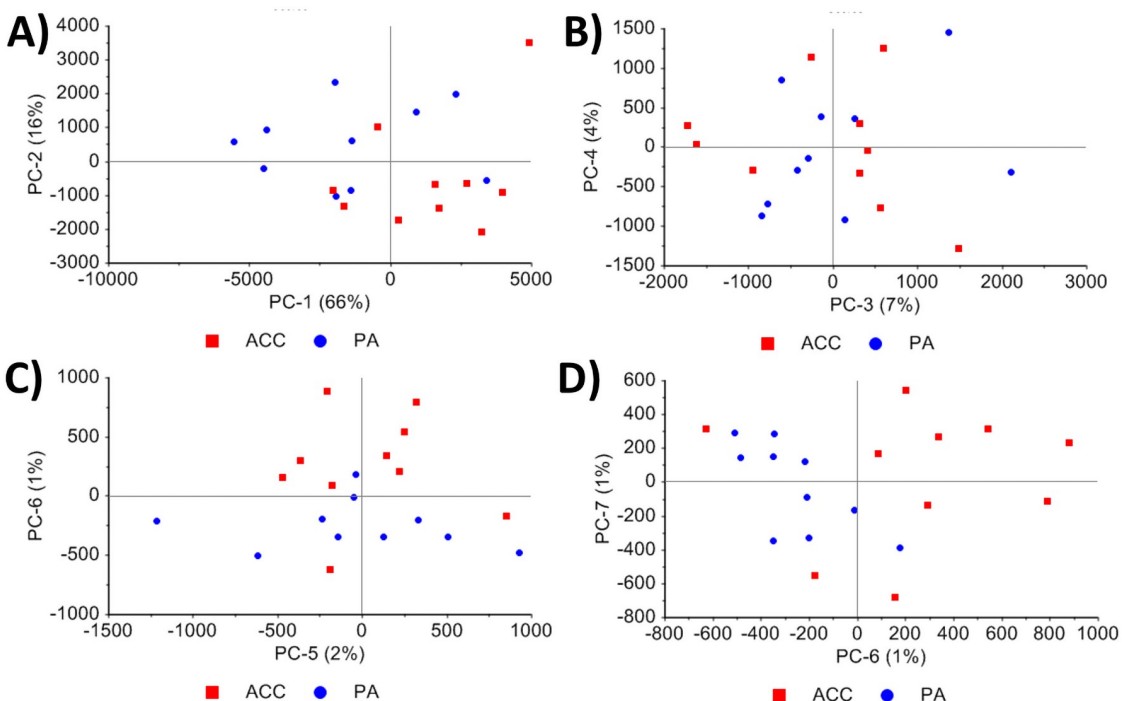

**Fig 4. Scores plots of the PCA: The scores of the A) PC-1 + PC-2, B) PC-3 + PC-4, C) PC-5 + PC-6 and D) PC-6 +PC-7 are illustrated.** The best data separation dependent on the tumor entity are seen by the PC-2 and PC-6.

The scores plots of PCA showed that the tumor entities PA and ACC separated on PC-2 (16%) and PC-6 (1%) (Figs 4 and 5). The loadings indicate, which spectral wavenumber ranges have the highest impact on clustering of the score values. Thus, the highest positive and negative loading values of PC-2 and PC-6 were identified (all peaks > 0.1 and < 0.1) and the PCA was recalculated for these spectral ranges. On PC-2 the positive loadings peaks from 1470–1454, 1135–1122, 1063–1054, and 1003–999 cm$^{-1}$ and the negative loading peaks from 1429–1421, and 989–985 cm$^{-1}$ were relevant for group-specific score value separation. On PC-6 the positive loadings peaks at 1289–1285, 1063–1042 cm$^{-1}$ and the negative loading peaks at 933–895 cm$^{-1}$ were relevant (Fig 5 and Table 3). By recalculation with the identified peaks of PC-2 and PC-6 a good data separation dependent on the tumor entity is possible (S2 Fig).

### Raman data analysis by PCA-LDA

In the PCA, PC-2 and PC-6 have been identified as most relevant to discriminate PA from ACC samples. Using PCA-LDA based on PC-1 to PC-6, a high overall accuracy for entity assignment of 90% was reached. Nine out of ten PA samples and nine out of ten ACC samples have been correctly classified (Table 4). Within this model, both the sensitivity and specificity were 90%. Matthew correlation coefficient and rigidity were 0.80 and 0.89, respectively.

### Discussion

One of the major issues in salivary gland tumor surgery is the pre- or intraoperative assessment of the tumor entity, particularly the differentiation between benign and malignant epithelial salivary gland neoplasms. Specific tumor type determination can be often challenging, sometimes it is impossible due to the broad histomorphologic variations in these tumors and the

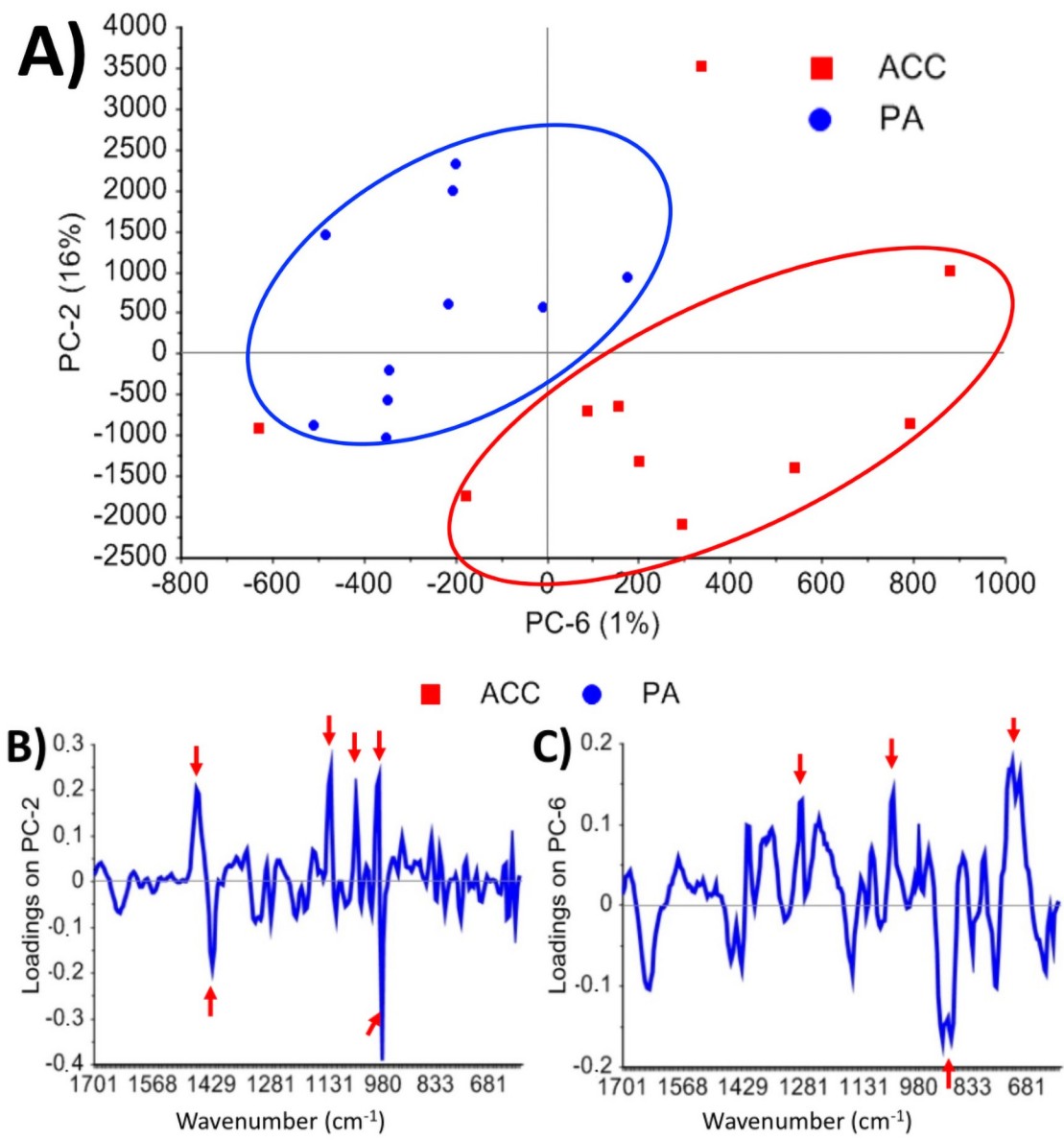

**Fig 5. Results of PCA.** A) shows the scores plot of the PCA in the spectral range from 600–1700 cm⁻¹. The tumor entities PA and ACC are well-separated by PC-2 and PC-6. B) and C) show the loadings plots of PC-2 and PC-6, respectively. Red arrows mark the peaks that have been chosen for recalculation of PCA.

importance of analyzing the interface of the tumor and the surrounding tissue, i.e. the identification of an infiltrative growth for the exact classification of some entities. Even for the here investigated common and comparatively well-defined salivary gland entities, PA and ACC, there can be several issues that may impede a correct diagnosis, particularly using FNAC, but also in histological samples [21, 22]. Thus, many attempts have been made to further delineate epithelial salivary gland neoplasms by application of immunohistochemical and molecular markers [23–25], but there are still many cases harboring diagnostic ambiguities. Therefore, this study focused on accessing the suitability of RS for salivary gland tumor entity assignment on the example of PA and ACC, as another diagnostic adjunct for the complex group of salivary gland neoplasms.

**Table 3. Raman wavenumber ranges (based on PCA) that are relevant to discriminate the tumor entity and major biochemical assignments (based on [14, 15]).**

| Wavenumber range [cm$^{-1}$] | Major biochemical assignments | Responsible for discrimination |
|---|---|---|
| 1470–1454 | Nucleic acid content, deoxyribose | PC-2 |
| 1429–1421 | Deoxyribose, lipids, DNA, RNA | PC-2 |
| 1289–1285 | Cytosine, nucleic acids | PC-6 |
| 1135–1122 | Proteins, fatty acids, 1126 cm$^{-1}$: paraffin | PC-2 |
| 1063–1054 | DNA, RNA, lipids | PC-2, PC-6 |
| 1003–999 | Phenylalanine | PC-2 |
| 989–985 | not defined | PC-2 |
| 933–895 | Saccharides, collagens, 907 cm$^{-1}$: formalin | PC-6 |
| 745–706 | DNA, RNA, lipids | PC-6 |

Previously published results by Yan *et al.* and Brozek-Pluska *et al.* focused on the differentiation of SGT tissue and salivary gland non-tumor tissue by the use of Raman spectroscopy [16, 26]. However, in contrast to other tumor locations in head and neck oncology, the main challenge in SGT surgery is not to detect the tumor borders in order to obtain clear and oncological safe margins without resecting too much healthy tissue. Instead of that, the precise pre-/ intraoperative determination of the tumor entity and thereby the best suitable operative strategy are unanswered issues. Therefore, the present study focused on assessing the value of RS for SGT entity assignment using the example of PA and ACC as model entity for benign and malignant SGT. The aim of this pilot project was to provide a proof-of-principle, that RS is applicable in SGT assessment in principle. This assessment included an evaluation of the material handling, of the required timeline of such measurements, of the RS-related biological variability and of the technical prerequisites. Based on the results of our study, further experiments with larger included case series are necessary in order to obtain statistically valuable statements.

We could detect higher Raman intensities for biochemical molecular vibrational structures that assign to nucleic acids, lipids, amides, and phenylalanine in ACC. Higher peak intensities were found in wavenumber ranges that code for proteins, collagen and saccharides in PA. To our best knowledge, this is the first RS-based study that includes ACCs.

Higher content of nucleic acids, amides, and phenylalanine in the malignant ACC compared to the benign PA are well allegeable. Comparing the finding of a higher lipid contend of ACC compared to PA, Yan *et al.* described an increased lipid content of PA compared to the healthy parotid gland tissue [17]. In another study, Malini and colleagues examined the potential of RS in terms of distinguishing between normal, inflammatory, pre- and malignant tissues. By analyzing the Raman spectra, they were able to determine differences in the various tissue alterations. In addition, they found a decrease of lipid RS signal in tissue parts, which were attributed to pathological conditions of the oral mucosa [27]. However, all to the authors' best knowledge, published articles focus on comparing healthy to tumor parotid gland tissue in the RS peak assignment. By performing this study on deparaffinized sections, especially the

**Table 4. Confusion matrix of PCA-LDA model with an overall accuracy of 90%.**

| | | True labels | |
|---|---|---|---|
| | | PA | ACC |
| Predicted labels | PA | 9 | 1 |
| | ACC | 1 | 9 |
| Accuracy | | 90% | 90% |

lipid fraction can be altered compared to the native tissue. Therefore, the results in regard to the lipid content of the SGT have only a limited significance.

Brozek-Pluska *et al.* could show, that RS can differentiate between tumor and non-tumor salivary gland tissue on the base of lipid and protein vibrations [26]. In 2011 Yan *et al.* found an increased Raman intensity for wavenumbers that code for proteins, lipids and DNA in PA samples compared to normal parotid gland tissue [16]. In a subsequent study, Yan *et al.* reported about a discrimination model of PA, Warthin's tumor, and mucoepidermoid carcinoma based on support vector machine algorithms for Raman-spectra data analysis [17]. However, they performed the mean spectra peak assignment only in comparison of spectra that were acquired in the tumor tissue versus spectra, that were acquired in normal salivary gland tissue [17]. There was no peak assignment comparing the different tumor entities [17].

Differences in tissue preparation and fixation may influence the results of RS and must therefore be taken into account in the assessment. Remaining formalin and paraffin residues typically induce enlarged Raman intensities around 907 and 1126 cm$^{-1}$, respectively [14]. As the influence of formalin fixation on the Raman spectral intensities is discussed to be negligible [28, 29], the process of paraffinization and deparaffinization may have a higher influence on Raman spectral intensities in terms of remaining paraffin. Furthermore, the deparaffinization procedure leads to changes in the remaining tissue by washing out especially the lipid fraction [29–31]. We addressed this limitation by using as unified deparaffinizing conditions as possible. However, one explanation for the low impact of wavenumber areas, which are assigned to lipids, could be the fact, that our study was performed using deparaffinized tissue instead of fresh tissue [30].

To reflect tumor heterogeneity *in vivo*, the measurements were taken randomly in nuclei, cytoplasm, and extracellular matrix. We were aware that this could lead to a less accurate classification since others showed that Raman spectra of cell nuclei, cytoplasm, and ECM significantly differ from each other [32]. The tumor tissue heterogeneity could be one reason for the high standard deviation values in the entity specific mean spectra. By performing a PCA with all single acquired spectra, we observed a clustering dependent at the single samples. These observations implicate, that there is a sample-depended Raman spectroscopy signature, besides of the tissue heterogeneity.

We aimed to find out if despite such potential variance and random measurement RS was suitable to extract tumor entity-specific information. Most important, it has to be ensured that RS measurements are reliably performed on tumor cells and not in non-tumor tissue. However, intraoperative biopsies might contain both, tumor and normal tissue. In such cases, it is mandatory to clearly identify the tumor within the surrounding healthy tissue in order to measure within the tumor exclusively. To the opinion of the authors, it will be necessary to identify the tumor tissue by for example the use of a classical HE histology on a consecutive cut. HE staining is also possible in frozen sections. This is a reliable way to identify the tumor tissue and exclude non-tumor tissue parts. Falsely measured non-tumor tissue would increase the variance of RS.

Moreover, the theoretically high accuracy of 90% in the PCA-LDA-model for discriminating PA and ACC, based on well explainable differences in the Raman sensitive biochemical molecular vibrational mode, makes the RS a promising additive tool for improvement of entity assignment in preoperative cytology diagnostics and intraoperative frozen section. The authors are aware of the fact, that limited case numbers like in our study are not suitable to reliably determine the accuracy of a novel diagnostic procedure. However, the clearly visible differences between our results indicate that RS seems to be able to ameliorate SGT diagnostics in the future. In order to further assess whether RS is suitable for determining the entity of SGT, both, the inclusion of more tumor entities and the prospective review of the multivariate

algorithm is necessary. RS is based on a different technological approach than the used standard techniques for SGT entity determination of histology and immunohistology. Therefore, RS could potentially help to differentiate SGT entities, which are difficult to discriminate by histological-morphology and immunohistology criteria.

In order to obtain spectroscopic measurement results based on the molecular tissue composition, as many artificial influencing factors as possible must be controlled. Therefore, uniformly tissue preparation and fixation protocols, measurement protocols and a technical RS measurement setup that allows an accurate identification and measurement of the tumor tissue by excluding the impact of non-tumor tissue and avoid the influence of artificial materials like the cover glass, are necessary. Based on the results of this study and in respect to existing workflows in the pathological diagnostic, the most likely tissue sample preparation is cutting in a frozen section. However, by tissue preparation in a frozen section, there is a standardization of the conditions like tissue thickness possible. Current studies of our group address the question of the influence of formalin and paraffin fixation on the biological spectral information. Furthermore, the influence of the different cover slide materials is analyzed in detail as well. We did not compare SGT tissue to healthy salivary gland tissue. However, the proof that such differentiation seems to be possible was already provided by other groups [16, 17, 26].

## Conclusion

The peak assignment of the entity-specific mean spectra showed that spectra measured in ACC were dominated by nucleic acids, lipids, and amides and in PA by collagens, and saccharides. Furthermore, by the use of PCA the relevant peaks for entity discrimination were identified. Based on a PCA-LDA model an entity determination with an overall accuracy of 90% was yielded.

To achieve reliable RS data acquisition and processing, all factors, which could have an influence on the biological relevant spectral information, have to be minimized. Therefore, tissue preparation and fixations protocols, measurement protocols, the technical RS measurement setup and data processing have to be standardized.

RS is a promising technology that could help to improve the entity determination of SGT in future. Only small amounts of tissue are necessary for RS measurements.

## Supporting information

**S1 Fig. Scores clustering dependent on the individual tumor samples: Showing the scores of PC-2 and PC-3, there is a clustering in dependence of the different tumor samples visible.**
(DOCX)

**S2 Fig. PCA-recalculation with the PC-2 and PC-6 high and low peaks: Scores plot after recalculation with the loading peaks bigger than 0.1 and smaller than -0.1 of the PC-2 and PC-6.** A good separation in dependence of the tumor entity is possible.
(DOCX)

**S1 Table. Mean spectra data of ACC and PA.**
(XLSX)

## Acknowledgments

We thank Heike Oberwinkler and Agnes Mörth for the great technical support.

## Author Contributions

**Conceptualization:** Till Jasper Meyer, Elena Gerhard-Hartmann, Maria Steinke, Stephan Hackenberg.

**Data curation:** Till Jasper Meyer, Elena Gerhard-Hartmann, Nina Lodes, Agmal Scherzad, Maria Steinke.

**Formal analysis:** Maria Steinke, Stephan Hackenberg.

**Funding acquisition:** Till Jasper Meyer, Stephan Hackenberg.

**Methodology:** Agmal Scherzad, Maria Steinke.

**Project administration:** Till Jasper Meyer, Agmal Scherzad, Stephan Hackenberg.

**Resources:** Maria Steinke.

**Supervision:** Rudolf Hagen.

**Validation:** Stephan Hackenberg.

**Writing – original draft:** Till Jasper Meyer, Maria Steinke, Stephan Hackenberg.

**Writing – review & editing:** Elena Gerhard-Hartmann, Nina Lodes, Agmal Scherzad, Rudolf Hagen.

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
