## [Decision Letter · Decision Letter 0]

31 May 2021

PONE-D-21-14279

Pilot study on the value of Raman spectroscopy in the entity assignment of salivary gland tumors

PLOS ONE

Dear Dr. Meyer,

Thank you for submitting your manuscript to PLOS ONE. After careful consideration, we feel that it has merit but does not fully meet PLOS ONE’s publication criteria as it currently stands. Therefore, we invite you to submit a revised version of the manuscript that addresses the points raised during the review process.

We look forward to receiving your revised manuscript.

Kind regards,

Yihong Wang

Academic Editor

PLOS ONE

Additional Editor Comments:

As the Academic Editor, I have a couple of general suggestions. First, please respect the reviewers by taking their suggestions seriously. Second, if you decide to revise and resubmit the manuscript, please address how you followed the comments/recommendations and why you did not follow the suggestions. Please include specific citations to justify your responses if you disagree with suggestions from the reviewers.

Journal Requirements:

2. In your ethics statement in the manuscript and in the online submission form, please ensure that you have discussed whether all data/samples were fully anonymized before you accessed them and/or whether the IRB or ethics committee waived the requirement for informed consent. If patients provided informed written consent to have data/samples from their medical records used in research, please include this information.

Reviewers' comments:

Reviewer's Responses to Questions

**Comments to the Author**

1. Is the manuscript technically sound, and do the data support the conclusions?

Reviewer #1: Yes

Reviewer #2: Partly

Reviewer #3: Partly

Reviewer #4: Partly

2. Has the statistical analysis been performed appropriately and rigorously? 

Reviewer #1: Yes

Reviewer #2: No

Reviewer #3: Yes

Reviewer #4: Yes

3. Have the authors made all data underlying the findings in their manuscript fully available?

Reviewer #1: Yes

Reviewer #2: No

Reviewer #3: No

Reviewer #4: No

4. Is the manuscript presented in an intelligible fashion and written in standard English?

Reviewer #1: No

Reviewer #2: Yes

Reviewer #3: Yes

Reviewer #4: Yes

5. Review Comments to the Author

Reviewer #1: The aim of this study is to evaluate the suitability of Raman spectroscopy for entity assignment in salivary gland tumors. In a principal component based linear discriminant analysis, majority of tumor samples are classified correctly. The topic of this paper is meaningful and interesting. However, there are still some concerns need to be addressed.

1. The readability of the article needs to be improved. The proportion of text narratives seems to be somewhat large. It is recommended that the authors simplify the text and add some charts to vividly describe the phenomenon revealed by the data.

2. As mentioned in the previous comment, it is recommended that authors supplement a flowchart and a mind map to better present their paper.

3. The resolutions of the figures in this paper need to be improved.

4. At the bottom of the Introduction, the authors need to summarize the remainder of the paper.

5. In the Materials and Methods section, the authors could add a pipeline diagram for tissue sampling and data processing to facilitate readers to better understand the paper.

6. In Raman spectroscopy section, a figure illustrating the tumor-specific areas may be helpful and a comparison of a healthy area also needs to be provided.

7. The authors need to add a separate section to briefly introduce the principle of Raman measurement and the principle of measuring instrument. At this time, it would be better to provide some schematic diagrams.

8. More details of the seven principal components should be given.

9. The experimental results and discussions need to be enriched.

Reviewer #2: The authors applied Raman spectroscopy to discriminate pleomorphic adenomas (PA) and adenoid cystic carcinomas (ACC) for assessment of salivary gland tumors. Although Raman spectra of rare samples are very valuable, this reviewer thinks that the authors need to explain and evaluate properly their methods and conditions of the analysis.

The authors seem to estimate the accuracy of their discrimination model according to the results of test datasets that are used to build the discrimination model itself. The reliability of the analytical model must be evaluated with the totally independent datasets. The number of spectra may not be enough to keep the test and validation datasets independently, because the sample is too rare. In such a case, leave-one-out cross validation method should be applied at least.

The authors must describe spectral treatment in detail. The spectra in Fig. 2 seems to be processed by baseline correction and/or background subtraction. The background spectra could give effect to the results of multivariate analysis.

This reviewer does not understand how the authors avoid the noise arising from paraffin. The authors describe “We addressed this limitation by using as unified deparaffinizing conditions as possible.” Commercial paraffine products are usually consist of multiple paraffine species. They have different characters in adsorption to the materials in tissue, which often give problems in erasing paraffine noise in the Raman analysis of paraffine fixed tissues. The authors must explain in detail.

The PCA seems to be applied to the averaged spectra consisting of 30 spectra each. However, the spectra of tumor-free areas, nerves, muscles, and blood vessels are excluded. The authors described that one of the major issues is the pre- or intraoperative determination of the tumor entity. Is it feasible to avoid those tissues visually without HE staining? How large it the focus volume of the Raman microscope? Do authors think that use of Raman probe which usually large focus volume is not suitable?

The spectra should include a band near 1740 cm-1 due to the C=O stretching mode to distinguish fat and lipid.

Reviewer #3: In this contribution, the authors present a pilot study to use Raman spectroscopy for the entity determination in salivary gland tumors. Raman spectroscopy from de-paraffinized sections of pleomorphic adenomas (PA) and adenoid cystic carcinomas (ACC) samples (10 each) were measured and analyzed using principle component analysis (PCA). Classification accuracy of about 90% was achieved separating the spectrum from the two groups. While interesting, the current study has several major limitations, which needs to be properly addressed.

1. The clinical significance of the proposed Raman spectroscopy measurements is not clear or convincing. The authors tried to make a case for speed and use Raman “for quick intraoperative diagnosis or to improve the FNAC accuracy.” However, the argument is not convincing, as the samples need to be sectioned and Raman measurements are not fast. The experiments conducted were using paraffin fixed samples and does not reflect the proposed clinical use scenarios. The authors are suggested to use fresh or frozen section samples to demonstrate the feasibility.

2. The number of specimens used for each group is very limited (10 each). Also, there is no control groups from healthy salivary gland samples. The authors acknowledged these limitations, but did not include more samples to address these issues.

3. Tissue heterogeneity may also affect the measurements and classification accuracy. How to better control this in clinical settings? The authors mentioned to use H&E slide to select only the tumor regions for analysis. But what about “quick intraoperative diagnosis” if H&E still needs to be prepared?

4. Can images be generated from the Raman measurements and co-registered with histology or immunohistology slides? They may be more informative.

Reviewer #4: The article is interesting and generally well written. The authors identify their limitation of not having normal salivary glad tissue characteristics in the model which could diminish the importance of their research work. If the reason of not having a normal tissue group is due to lack of samples, it's better to mention that in the article, or else if you have acquired samples after the manuscript submission then you could include those in the revised manuscript. My major concern with the article is that the authors mention that they collected 30 Raman spectra per sample which is reasonable for a pilot study and they have 20 samples in the study which should add up to 300 spectra for each group in the Figure 3 PCA plot but I don't see that. Did you average the spectra into one spectrum before doing the statistical analysis? If so, that will mask the spread of the data, the heterogeneity of the tissue that would be important to readers to see how each spectrum behave in the PCA space and understand it. The sample size is limited if you average by the sample and then do the analysis, unless you add a lot more samples, the conclusions drawn from the study may not be meaningful.

6. PLOS authors have the option to publish the peer review history of their article (what does this mean?). If published, this will include your full peer review and any attached files.

Reviewer #1: No

Reviewer #2: No

Reviewer #3: No

Reviewer #4: No

---

## [Author Response · Author response to Decision Letter 0]

23 Jul 2021

Reviewer #1: The aim of this study is to evaluate the suitability of Raman spectroscopy for entity assignment in salivary gland tumors. In a principal component based linear discrimi-nant analysis, majority of tumor samples are classified correctly. The topic of this paper is meaningful and interesting. However, there are still some concerns need to be addressed.

1. The readability of the article needs to be improved. The proportion of text narratives seems to be somewhat large. It is recommended that the authors simplify the text and add some charts to vividly describe the phenomenon revealed by the data.

Thank you for catching this point. We revised the text passages and added some.

2. As mentioned in the previous comment, it is recommended that authors supplement a flowchart and a mind map to better present their paper.

Thank you for this further idea. We added a flow chart, that presents the tissue preparation steps.

3. The resolutions of the figures in this paper need to be improved.

We have enlarged the resolution of all figures.

4. At the bottom of the Introduction, the authors need to summarize the remainder of the paper.

We would like to address this concern. However to our best knowledge the submission guide-lines of PLOS ONE does not contain a “remainder”.

5. In the Materials and Methods section, the authors could add a pipeline diagram for tissue sampling and data processing to facilitate readers to better understand the paper.

We added a diagram for the description of the tissue sampling to the manuscript (figure 1).

6. In Raman spectroscopy section, a figure illustrating the tumor-specific areas may be help-ful and a comparison of a healthy area also needs to be provided.

This information is included in the new figure 1.

7. The authors need to add a separate section to briefly introduce the principle of Raman measurement and the principle of measuring instrument. At this time, it would be better to provide some schematic diagrams.

Thank you for catching this point. We added a section to the introduction and the new figure 1.

8. More details of the seven principal components should be given.

We added more information about all the different principal components in the new figure 4.

9. The experimental results and discussions need to be enriched.

According to point 8 we added additional results and also added more details to the discus-sion. Furthermore the recalculation data with the high peaks of PC-2 and PC-6 and a PCA with all single spectra were added for enrichment of the results.

Reviewer #2: The authors applied Raman spectroscopy to discriminate pleomorphic adeno-mas (PA) and adenoid cystic carcinomas (ACC) for assessment of salivary gland tumors. Alt-hough Raman spectra of rare samples are very valuable, this reviewer thinks that the authors need to explain and evaluate properly their methods and conditions of the analysis.

We addressed this concern by adding new figures, which show in detail, who the tissue pro-cessing and Raman measurement procedure.

The authors seem to estimate the accuracy of their discrimination model according to the results of test datasets that are used to build the discrimination model itself. The reliability of the analytical model must be evaluated with the totally independent datasets. The number of spectra may not be enough to keep the test and validation datasets independently, because the sample is too rare. In such a case, leave-one-out cross validation method should be ap-plied at least.

For leave-one-out cross validation we calculated 20 different LDA-models with leaving every time one sample out. The left out sample we predicted with the remaining data set. The over-all accuracy of the model was 80%.

Confusion matrix of LOO-PCA-LDA model with an overall accuracy of 80%.

 True labels

 PA ACC

Predicted labels PA 9 1

 ACC 3 7

The authors must describe spectral treatment in detail. The spectra in Fig. 2 seems to be pro-cessed by baseline correction and/or background subtraction. The background spectra could give effect to the results of multivariate analysis.

In deed baseline correction was performed by the use of the customized software of the Ra-man spectroscopy system. We added this information to the method section. Thank you for this valuable advice.

This reviewer does not understand how the authors avoid the noise arising from paraffin. The authors describe “We addressed this limitation by using as unified deparaffinizing conditions as possible.” Commercial paraffine products are usually consist of multiple paraffine species. They have different characters in adsorption to the materials in tissue, which often give prob-lems in erasing paraffine noise in the Raman analysis of paraffine fixed tissues. The authors must explain in detail.

We addressed this by the use of a deparaffination protocol. However there will be still a rest of paraffin and furthermore especially fat fraction get washed away. But all samples were treated with the same fixation and deparaffination protocol, so that the conditions in term of paraffin-contamination should be comparable between the different samples. To our point of view the question in which way the biological Raman spectral information of the tissue is in-fluenced by paraffination and deparaffination is important for development of RS applica-tions in the future. Therefore we will address this question in detail in a upcoming project.

The PCA seems to be applied to the averaged spectra consisting of 30 spectra each. However, the spectra of tumor-free areas, nerves, muscles, and blood vessels are excluded. The authors described that one of the major issues is the pre- or intraoperative determination of the tu-mor entity. Is it feasible to avoid those tissues visually without HE staining? How large it the focus volume of the Raman microscope? Do authors think that use of Raman probe which usually large focus volume is not suitable?

To our point of view the tumor identification in HE-sections is definitely important. Therefore the co-registration of histology and spectroscopic evaluated slides is absolutely necessary and was performed in our study. HE-staining is also for frozen sections possible. Therefore we evaluate in a current study the use of consecutive frozen sections, native for spectroscopic measurements and HE stained for tumor identification. Only by histological tumor identifica-tion is a reliable tumor identification possible. There are research groups, that publish the use of Raman spectroscopy with a large focus volume, which was directly applicated to the mac-roscopic identified tumor surface. However to the authors opinion this approach is interesting, but the missing micromorphological control makes the accidental measurement of non-tumor structures more likely. This will enlarge the variation and therefore is the further development to a clinical translation more unlikely. To the authors view a translation to clinic of spectro-scopic approaches is only realistic by the use in addition to the well-established classical his-topathological analysis.

The spectra should include a band near 1740 cm-1 due to the C=O stretching mode to dis-tinguish fat and lipid.

Indeed there is a discussion about the size of the biological relevant wavenumber range. It is common to include the wavenumber range from 1800 cm-1. We recalculated with the wave-number range from 1800-600 cm-1. Including the wavenumber range 1800-1700 cm-1 had no impact on the model accuracy.

Reviewer #3: In this contribution, the authors present a pilot study to use Raman spectros-copy for the entity determination in salivary gland tumors. Raman spectroscopy from de-paraffinized sections of pleomorphic adenomas (PA) and adenoid cystic carcinomas (ACC) samples (10 each) were measured and analyzed using principle component analysis (PCA). Classification accuracy of about 90% was achieved separating the spectrum from the two groups. While interesting, the current study has several major limitations, which needs to be properly addressed.

1. The clinical significance of the proposed Raman spectroscopy measurements is not clear or convincing. The authors tried to make a case for speed and use Raman “for quick intraopera-tive diagnosis or to improve the FNAC accuracy.” However, the argument is not convincing, as the samples need to be sectioned and Raman measurements are not fast. The experiments conducted were using paraffin fixed samples and does not reflect the proposed clinical use scenarios. The authors are suggested to use fresh or frozen section samples to demonstrate the feasibility.

In deed the chosen experimental design by using deparaffinized tissue does not reflect the pursued setting to 100 %. However for a first evaluation, if RS is principally suitable for sali-vary gland tumor entity determination, we consciously have chosen deparaffinized tissue samples. The main reason is, that malignant salivary gland tumors are too rare to collect enough tissue samples in an adequate time, also in a big tertiary universal hospital with a very high case load. Therefore there is to our point of view no other way, to address this im-portant research question in rarer diseases like salivary gland cancer.

2. The number of specimens used for each group is very limited (10 each). Also, there is no control groups from healthy salivary gland samples. The authors acknowledged these limita-tions, but did not include more samples to address these issues.

The main reason for the limited number of specimens is listed above. Usually spectroscopy approaches in cancer science were used for cancer detection. But in salivary gland tumor treatment is opposed to the classical head and neck squamous cell carcinoma treatment, the main challenge the entity and dignity determination but not the cancer border detection. Therefore in this clinical context healthy salivary gland tissue is not the relevant control group.

3. Tissue heterogeneity may also affect the measurements and classification accuracy. How to better control this in clinical settings? The authors mentioned to use H&E slide to select only the tumor regions for analysis. But what about “quick intraoperative diagnosis” if H&E still needs to be prepared?

Thank you for catching this point. HE staining is also in frozen sections possible. In a future study we address this problem by using consecutive frozen sections, native for spectroscopic measurements and HE stained for tumor identification. To the authors point of view the use of RS for in-vivo measurement is unrealistic, because of so many uncontrollable variables. The chance of translation innovative methods like RS to the clinic is much more probably in sup-porting existing and standardized clinical procedures like intraoperative frozen sections.

4. Can images be generated from the Raman measurements and co-registered with histology or immunohistology slides? They may be more informative.

To our point of view the tumor identification in HE-sections is definitely important. Therefore the co-registration of histology and spectroscopic evaluated slides is absolutely necessary and was performed in our study.

Reviewer #4: The article is interesting and generally well written. The authors identify their limitation of not having normal salivary glad tissue characteristics in the model which could diminish the importance of their research work. If the reason of not having a normal tissue group is due to lack of samples, it's better to mention that in the article, or else if you have acquired samples after the manuscript submission then you could include those in the re-vised manuscript. My major concern with the article is that the authors mention that they collected 30 Raman spectra per sample which is reasonable for a pilot study and they have 20 samples in the study which should add up to 300 spectra for each group in the Figure 3 PCA plot but I don't see that. Did you average the spectra into one spectrum before doing the statistical analysis? If so, that will mask the spread of the data, the heterogeneity of the tissue that would be important to readers to see how each spectrum behave in the PCA space and understand it. The sample size is limited if you average by the sample and then do the analy-sis, unless you add a lot more samples, the conclusions drawn from the study may not be meaningful.

Thank you for catching these important points. In deed we used average spectra for the mod-el calculation. The reason for that approach is, that by building the average spectra a kind of spectra normalization is possible. To address your concern we added the PCA with the single spectra (see S1). The labeling is dependent of the sample number. A clustering along the sample number is visible. 

Due to the rarely incidence of malignant salivary gland tumors, an inclusion of many more samples is not possible, also in a big tertiary high volume universal hospital.

---

## [Decision Letter · Decision Letter 1]

4 Aug 2021

PONE-D-21-14279R1

Pilot study on the value of Raman spectroscopy in the entity assignment of salivary gland tumors

PLOS ONE

Dear Dr. Meyer,

Thank you for submitting your manuscript to PLOS ONE. After careful consideration, we feel that it has merit but does not fully meet PLOS ONE’s publication criteria as it currently stands. Therefore, we invite you to submit a revised version of the manuscript that addresses the points raised during the review process.

We look forward to receiving your revised manuscript.

Kind regards,

Yihong Wang

Academic Editor

PLOS ONE

Journal Requirements:

Additional Editor Comments:

The authors had satisfactorily addressed most of the reviewers’ comments. Please revise the manuscript and address additional questions/comments by reviewers.

Reviewers' comments:

Reviewer's Responses to Questions

**Comments to the Author**

1. If the authors have adequately addressed your comments raised in a previous round of review and you feel that this manuscript is now acceptable for publication, you may indicate that here to bypass the “Comments to the Author” section, enter your conflict of interest statement in the “Confidential to Editor” section, and submit your "Accept" recommendation.

Reviewer #1: (No Response)

Reviewer #2: (No Response)

Reviewer #3: All comments have been addressed

2. Is the manuscript technically sound, and do the data support the conclusions?

Reviewer #1: Partly

Reviewer #2: No

Reviewer #3: Yes

3. Has the statistical analysis been performed appropriately and rigorously? 

Reviewer #1: N/A

Reviewer #2: No

Reviewer #3: Yes

4. Have the authors made all data underlying the findings in their manuscript fully available?

Reviewer #1: Yes

Reviewer #2: Yes

Reviewer #3: Yes

5. Is the manuscript presented in an intelligible fashion and written in standard English?

Reviewer #1: No

Reviewer #2: No

Reviewer #3: Yes

6. Review Comments to the Author

Reviewer #1: Some of my concerns have been tackled, but I am still unsatisfied with the presentation and solidity of this manuscript. Here are some remaining questions in the last review and new concerns.

1. The presentation of this manuscript is improvable.

- The proportion of text narratives still seems to be large.

- The resolutions of the figures can be improved further.

2. This paper is not solid in theory and the experimental results are failed to support their claim.

- The experimental results and discussions are not enough.

- As other reviewers said the clinical significance of the proposed Raman spectroscopy measurements is not convincing.

- The sample size is not enough to support the results.

Reviewer #2: The authors have not improved their manuscript sufficiently. It includes lots of inaccurate results and discussions. Although their data is valuable, the manuscript must be revised well before publishing.

The authors must describe how they made the baseline correction in detail and validate correctness of the process. Especially in multivariate analysis of Raman spectra, one must pay a big attention to the selection of background spectra. If one used 2 background spectra for correction of datasets and the datasets were categorized into 2 groups, the result could merely reflect the patterns of the background spectra. The authors must carefully explain how they avoid possible mistakes.

The PCA seems to be applied to the averaged spectra consisting of 30 spectra each. Their discrimination model could be applied only to the averaged spectrum of 30 spectra and could not be applicable to analysis of each Raman spectrum. Consequently, the value of the accuracy of their model, 90% that the authors describe, does not correct for a single spectrum.

The accuracy of a discrimination model must be evaluated with an independent dataset. In case that the number of data is too few to keep some data out for validation, one may use leave-one-out cross-validation method. As the author describe in the rebuttal letter, the overall accuracy to the averaged 30 spectra is 80%. It is not 90%.

The bands in the spectra must be correctly labelled. In Table 2, a band at 1003-989 cm-1 is assigned to phenylalanine. However, phenylalanine always shows 2 bands near 1003 and 1030 cm-1. In contrast, fat species often show a sharp single band near 990 cm-1 and the spectra in Fig. 3 looks similar to their spectral pattern. The authors must show the spectral area from 1800 cm-1 to reveal their assignment.

Reviewer #3: (No Response)

7. PLOS authors have the option to publish the peer review history of their article (what does this mean?). If published, this will include your full peer review and any attached files.

Reviewer #1: No

Reviewer #2: No

Reviewer #3: No

---

## [Author Response · Author response to Decision Letter 1]

27 Aug 2021

Reviewer #1: Some of my concerns have been tackled, but I am still unsatisfied with the presentation and solidity of this manuscript. Here are some remaining questions in the last review and new concerns.

1. The presentation of this manuscript is improvable.

- The proportion of text narratives still seems to be large.

Thank you for this advice. We adjusted the manuscript and reduced the narratives.

- The resolutions of the figures can be improved further.

We have further improved the figure resolution and take care for the journal specific adjustments.

2. This paper is not solid in theory and the experimental results are failed to support their claim.

- The experimental results and discussions are not enough.

The claim of the manuscript is to basically proof that there is an entity dependent difference in in RS-signature. To our best knowledge, there is no other published manuscript that focusses on the question, if entity assignment in salivary gland cancer is in general possible. Our results clearly show that there is an entity dependent RS signature, focusing on differences in the mean spectra and multivariate data analysis. To our point of view we define important steps, that are really necessary to evaluate the possibility of clinical translation. Our study implements the RS technology and reliable proof of concept data with a clear focus on future clinical application, which is often neglected on other studies. 

- As other reviewers said the clinical significance of the proposed Raman spectroscopy measurements is not convincing.

We regret, that we can´t carve out the clinical significance of the proposed Raman spectroscopy measurements. We adjusted the introduction. The clinical significance of the problem with the entity assignment in salivary gland tumors is clearly obvious for all clinicians who are immediately involved in the surgery therapy of salivary gland cancer. However, the clinical significance should be very obvious also for non-clinicians after reading the manuscript. Therefore, we restructured the introduction. 

- The sample size is not enough to support the results.

Focusing on salivary gland cancer the incidence is low. However, also in a big tertiary care hospital much bigger number of cases is unrealistic. But to our point of view there is a need also to focus on tumor entities with low incidence. Comparing to other studies dealing with salivary gland cancer, the sample size is similar (e.g. Brozek-Pulska et al.). Please take in mind, that every patient is for us only one sample, a big difference to other manuscripts. If we focussed on our main tumor entities like squamous cell carcinoma a sample size of hundred and more would be managable.

Reviewer #2: The authors have not improved their manuscript sufficiently. It includes lots of inaccurate results and discussions. Although their data is valuable, the manuscript must be revised well before publishing.

The authors must describe how they made the baseline correction in detail and validate correctness of the process. Especially in multivariate analysis of Raman spectra, one must pay a big attention to the selection of background spectra. If one used 2 background spectra for correction of datasets and the datasets were categorized into 2 groups, the result could merely reflect the patterns of the background spectra. The authors must carefully explain how they avoid possible mistakes.

Thank you very much for addressing this point. Indeed, the baseline correction can have a big impact on the results. We have performed the baseline correction with the data-export for every single patient sample independently. Therefore, there should be no influence on later performed entity assignment within the multivariate data analysis.

The PCA seems to be applied to the averaged spectra consisting of 30 spectra each. Their discrimination model could be applied only to the averaged spectrum of 30 spectra and could not be applicable to analysis of each Raman spectrum. Consequently, the value of the accuracy of their model, 90% that the authors describe, does not correct for a single spectrum.

We totally agree with your statement. But it was not the aim, to validate the model for single spectra. Focusing on the clinical point of view, we used average spectra to reduce the power of single measurements respectively single spectra. We added this important point to the discussion.

The accuracy of a discrimination model must be evaluated with an independent dataset. In case that the number of data is too few to keep some data out for validation, one may use leave-one-out cross-validation method. As the author describe in the rebuttal letter, the overall accuracy to the averaged 30 spectra is 80%. It is not 90%.

Indeed, the highest level of validation is the use of an independent dataset. To our point of view, this should be the next step. We performed leave-one-out cross-validation. The overall accuracy is in this model 80% and therefore lower than 90% in the in the LDA model. However, the leave-one-out cross-validation confirmed, that in general an entity assignment based on Raman spectra data is possible.

The bands in the spectra must be correctly labelled. In Table 2, a band at 1003-989 cm-1 is assigned to phenylalanine. However, phenylalanine always shows 2 bands near 1003 and 1030 cm-1. In contrast, fat species often show a sharp single band near 990 cm-1 and the spectra in Fig. 3 looks similar to their spectral pattern. The authors must show the spectral area from 1800 cm-1 to reveal their assignment.

Thank you for this advice. We changed figure 3, to show the spectral area from 1800 cm-1 and corrected the labeling.

Reviewer #3: (No Response)

---

## [Editor Report · Decision Letter 2]

2 Sep 2021

Pilot study on the value of Raman spectroscopy in the entity assignment of salivary gland tumors

PONE-D-21-14279R2

Dear Dr. Meyer,

We’re pleased to inform you that your manuscript has been judged scientifically suitable for publication and will be formally accepted for publication once it meets all outstanding technical requirements.

Kind regards,

Yihong Wang

Academic Editor

PLOS ONE

Additional Editor Comments (optional):

All the comments have been addressed.
---

## [Editor Report · Acceptance letter]

8 Sep 2021

PONE-D-21-14279R2 

Pilot study on the value of Raman spectroscopy in the entity assignment of salivary gland tumors 

Dear Dr. Meyer:

I'm pleased to inform you that your manuscript has been deemed suitable for publication in PLOS ONE. Congratulations! Your manuscript is now with our production department. 

Kind regards, 

on behalf of

Dr. Yihong Wang 

Academic Editor

PLOS ONE